# A Simplified Multiplex PCR Assay for Simultaneous Detection of Six Viruses Infecting Diverse Chilli Species in India and Its Application in Field Diagnosis

**DOI:** 10.3390/pathogens12010006

**Published:** 2022-12-21

**Authors:** Oinam Priyoda Devi, Susheel Kumar Sharma, Keithellakpam Sanatombi, Konjengbam Sarda Devi, Neeta Pathaw, Subhra Saikat Roy, Ngathem Taibangnganbi Chanu, Rakesh Sanabam, Huirem Chandrajini Devi, Akoijam Ratankumar Singh, Virendra Kumar Baranwal

**Affiliations:** 1ICAR Research Complex for NEH Region, Manipur Centre, Lamphelpat, Imphal 795004, India; 2Department of Biotechnology, Manipur University, Canchipur, Imphal 795003, India; 3Advanced Centre for Plant Virology, Division of Plant Pathology, ICAR-Indian Agricultural Research Institute, New Delhi 110012, India

**Keywords:** multiplex PCR, capsicum chlorosis orthotospovirus, chilli veinal mottle virus, large cardamom chirke virus, cucumber mosaic virus, pepper mild mottle virus, chilli leaf curl virus, field validation

## Abstract

Chilli is infected by at least 65 viruses globally, with a mixed infection of multiple viruses leading to severe losses being a common occurrence. A simple diagnostic procedure that can identify multiple viruses at once is required to track their spread, initiate management measures and manage them using virus-free planting supplies. The present study, for the first time, reports a simplified and robust multiplex PCR (mPCR) assay for the simultaneous detection of five RNA viruses, capsicum chlorosis orthotospovirus (CaCV), chilli veinal mottle virus (ChiVMV), large cardamom chirke virus (LCCV), cucumber mosaic virus (CMV), and pepper mild mottle virus (PMMoV), and a DNA virus, chilli leaf curl virus (ChiLCV) infecting chilli. The developed mPCR employed six pairs of primer from the conserved coat protein (CP) region of the respective viruses. Different parameters viz., primer concentration (150–450 nM) and annealing temperature (50 °C), were optimized in order to achieve specific and sensitive amplification of the target viruses in a single reaction tube. The detection limit of the mPCR assay was 5.00 pg/µL to simultaneously detect all the target viruses in a single reaction, indicating a sufficient sensitivity of the developed assay. The developed assay showed high specificity and showed no cross-amplification. The multiplex PCR assay was validated using field samples collected across Northeast India. Interestingly, out of 61 samples collected across the northeastern states, only 22 samples (36%) were positive for single virus infection while 33 samples (54%) were positive for three or more viruses tested in mPCR, showing the widespread occurrence of mixed infection under field conditions. To the best of our knowledge, this is the first report on the development and field validation of the mPCR assay for six chilli viruses and will have application in routine virus indexing and virus management.

## 1. Introduction

Chilli (*Capsicum* sp.) is an important commercial spice crop cultivated throughout the globe [1,2] with an annual production of 4.25 million tonnes of both fresh and dry chillies [3]. India is a major producer and exporter of red chillies, producing around 1.7 million tonnes [4]. India is home to varied forms of chilli *viz., C. annuum, C. frutescens,* and *C. chinense* [5], of which *C. annuum* is the most widely cultivated chilli across the country. Although the chilli is grown in all the states and union territories of the country, the Northeastern Region (NER) of India, comprising of states of Manipur, Nagaland, Assam, Sikkim, and Mizoram, holds unique importance in terms of the cultivation of one of the hottest chilli landraces known as King Chilli / Umorok / Naga chilli (*C. chinense*). This, in addition to several other landraces of chilli like *C. frutescens* and *C. annuum*, are grown traditionally since for ages, their usage in different ethnic cuisines and ethno-pharmacological practices have contributed significantly to the agricultural economy and livelihood of farmers [6].

Viral diseases have always been a major constraint to chilli production and productivity worldwide [7,8,9,10]. At least 65 viruses have been reported to infect the chilli worldwide [11]; 24 among them occur under natural infection, and the rest occur under experimental conditions [12]. Around 19 different viruses are known to infect chilli peppers (*Capsicum* sp.) in the Indian subcontinent [13]. The NER of India has recorded a widespread prevalence and high incidence of viral diseases on all the cultivated chilli landraces (King chilli and bird’s eye chilli) due to the favourable environmental conditions prevailing throughout the growing seasons [7,14]. A thorough identification through the accurate detection of these viral diseases is the only means of carving out effective management measures. As such, the most widely prevalent viruses infecting different cultivars and landraces of chilli across India as well as in the NER of India are capsicum chlorosis orthotospovirus (CaCV) [15,16], pepper mild mottle virus (PMMoV) [17], chilli veinal mottle virus (ChiVMV) [7,14], cucumber mosaic virus (CMV) [18,19], chilli leaf curl virus (ChiLCV) [20], and newly reported large cardamom chirke virus (LCCV) [21].

The capsicum chlorosis orthotospovirus, a member of the genus *Tospovirus* in the family *Tospoviridae*, has emerged as a serious threat to chilli crops in India during the last 10 to 15 years, leading to considerable economic losses [15,22]. The segmented single-stranded tripartite RNA genomes of CaCV consist of 8.9 kb (L RNA), ~5 kb (M RNA), and 2.9 kb (S RNA) in size, respectively [23], and are transmitted by thrips as well as mechanical inoculation [15,24]. Pepper mild mottle virus is a member of the tobamovirus group, having positive-strand 6.3 kb RNA genome. The PMMoV is a serious threat to chilli cultivation worldwide [25,26] and is extremely resistant to physical and chemical agents. It has non-enveloped, rod-shaped virions that are seed transmitted [27] and cause significant losses both under protected and field cultivation [17].

Another important virus, the chilli veinal mottle virus, a member of the genus *Potyvirus* in the family *Potyviridae*, has a wide host range and causes serious yield losses to chilli production all over the globe. The ChiVMV has a 9.7 kb single-stranded positive-sense RNA genome with a 3′-terminal poly (A) tail, which is prominently distributed in the Asian region [28]. Its occurrence is more common in East Asian countries, causing serious losses in chilli production [29,30]. Its infection and prevalence on *C. chinense* and *C. annuum* grown in the NER of India has been recorded during recent years [7,14]. The ChiVMV is naturally transmitted by different species of aphid vectors in a non-persistent manner. The ChiVMV is transmitted to various hosts of family Solanaceae, such as eggplants, tobacco, and tomatoes in Asian continent [31,32,33], and is suspected to be responsible in bringing havoc in the chilli plantations in the NER, where low to moderate temperature conditions prevail. The cucumber mosaic virus, a cucumovirus, has global distribution and is recognized as one of the most serious viral diseases of crop plants with a very wide host range expanding to approximately 365 genera and at least 85 plant families [34]. The CMV has a linear positive-sense, tripartite single-stranded RNA molecule with a total size of ~8.6 kb packaged in separate particles as RNA 1 (~3.3 kb), RNA 2 (~3.0 kb), and RNA 3 (~2.2 kb) [34,35,36]. It is transmitted by aphids and is also seed-borne [34,37]. The infection of CMV in chilli often leads to losses of 28–40% in yield, and in severe infections yield losses may reach up to 100% if plants are infected at an early stage [38,39], with the plants producing none or only very few fruits, which are small in size [40]. Large cardamom chirke virus, a member of the genus *Macluravirus* in the family *Potyviridae*, which naturally infects large cardamom was recorded to infect *C. annuum* (landrace Dalle Khursani), a solanaceous crop, for the first time, exhibiting the cross-species transmission capability of macluraviruses [21,41]. The LCCV, a positive sense single stranded RNA virus, is vectored by aphids, *Rhopalosiphum maidis*, *Brachycaudus helichrysi*, and *Myzus persicae* in a non-persistent manner [42,43]. Its natural cross-transmission to chilli has raised concerns over a possible new outbreak in solanaceous crops. The chilli leaf curl virus is one of the most devastating viral pathogens, causing chilli leaf curl disease, which results in enormous economic losses worldwide, especially in the Indian subcontinent [11,44,45]. The ChiLCV is a monopartite begomovirus that has a DNA-A of 2.7 kb and associated betasatellite of 1.3 kb size. The ChiLCV is one of the most prevalent viruses in chilli crops, leading to severe leaf curl symptoms in the Indian subcontinent [44,46]. The emergence of pathogenically and genetically diverse chilli begomoviruses through recombination is common [47]. The ChiLCV is efficiently spread by the whitefly (*Bemisia tabaci*), which is abundant year-round in tropical and subtropical climates where a wide variety of hosts serve as reservoirs [20].

The increase in global international trade and the commercial movement of plant materials at a domestic and international scale during recent years, coupled with the climate disruptions, has led to the emergence and re-emergence of diverse viruses in chilli crop, which were previously either not reported or not present in the geographical areas under study [21]. This, coupled with the phenomenon of host shifts, or the frequent overlapping of host ranges of several viruses [21,48], for most of the plant species including *Capsicum* spp. leads to the infection of multiple viruses in a single plant species [49]. Synergism resulting from a mixed infection of multiple viruses belonging to different taxonomic groups has been reported to induce severe pathogenicity and symptoms in the chilli, leading to a drastic reduction of the expression of defense-related genes [50,51]. It is generally difficult to identify an individual infection based only on symptoms, as complex symptoms with increased virulence are often induced by the infection of different viruses. A prolonged mixed infection may lead to the emergence of new viruses resulting from the recombination of co-replicating viral genotypes [52,53].

This scenario highlights the importance of the development of a robust and proper diagnosis of associated viral agents, which is very crucial in disease management [54,55]. Different techniques including serological (enzyme-linked immunosorbent assay, ELISA), molecular (molecular hybridization and DNA amplification), and high-throughput sequencing (HTS) has been developed for the identification and detection of viruses and virus strains [55]. ELISA is still the popular technique in the routine detection and screening of plant samples for viruses, as it can be performed with little training using commercially available antibodies specific to a virus. However, antibody production is expensive, time-consuming, and unpredictable, and it cannot be designed to cope with viral variability [55]. Further, the multiplexing of the ELISA assay is possible to a limited extent in addition to the issues of sensitivity.

Although many diagnostic assays have been developed, the benchmark detection of plant viruses still relies on the use of PCR based methods. The detection and identification of various viruses or viral strains from a single assay simultaneously reduces the time and cost of the analysis [56], which is convenient for evaluating mixed infections in individual plants. The detection of individual viruses in a sample is mainly based on the spatial separation of multiple viruses (targets) [57]. The development and availability of multiplex detection methods capable of the simultaneous detection of more than one virus in a single reaction, therefore, contributes in the reduction of the cost of testing, increased efficiency, and amenability of its routine use. Techniques like multiplex PCR (mPCR) are a quick, reliable, and cost-effective method used frequently due to the high specificity, and the sensitivity, which can reduce the cost and time, as a single reaction is able to detect multiple viruses simultaneously, especially when large numbers of samples need to be tested with high specificity [56]. Keeping in view the predominant infection of multiple viruses in chilli and the need to have simplified diagnostic assay, the present study reports on the development and field validation of mPCR assay for the simultaneous detection of six viruses.

## 2. Materials and Methods

### 2.1. Plant and Virus Materials

Initially, the virus infected leaf samples of chilli plants were collected from different chilli growing locations in Manipur, India. Infected plant samples as well as healthy chilli plants were maintained in a glasshouse under insect-proof conditions. Leaf samples of both infected and healthy chilli plants were stored at −80 °C for later use. RT-PCR and PCR detection were used for the confirmation of the viruses in these samples.

### 2.2. Nucleic-acid Extraction and cDNA Synthesis

Total RNA was extracted from healthy and infected chilli leaves using an RNeasy^®^ Plant mini kit (Qiagen, Hilden, Germany) following manufacturer’s protocol. The first-strand cDNA was synthesised using an M-MLV Reverse Transcriptase kit (Promega, Madison, WI, USA). A DNeasy^®^ Plant Mini Kit (Qiagen, Hilden, Germany) was used for the extraction of the total DNA from healthy and infected chilli leaves according to the product’s manual. The PCR amplified products were purified with a gel extraction kit (Qiagen, Hilden, Germany), ligated into a pGEM-T Easy Vector (Promega, Madison, WI, USA), and cloned in *Escherichia coli* DH5-alpha (Takara Bio Inc., Kusatsu, Shiga, Japan). The positive clones were screened with colony PCR, and the recombinant plasmids were bi-directionally sequenced at a commercial facility (Eurofins Genomics India Pvt. Ltd., Bengaluru, India).

### 2.3. Primer Design

The specific primers of the nucleocapsid (NP) gene/coat protein (CP) gene of CaCV, ChiVMV, ChiLCV, LCCV, CMV, and PMMoV were designed by aligning the nucleotide sequences downloaded from the NCBI GenBank database using MEGA7 software (https://mega.software.informer.com/7.0/; accessed on and before 22 February 2019). The primers were designed to target genes of the respective viruses with distinct amplicon size for easier visualization. Precautions were taken to avoid any non-specific amplification and the self- and cross-complementarity of primer pairs. Multiple sets of specific primers (three sets of primers for ChiVMV, ChiLCV, and CMV and four sets of primers for LCCV, CaCV, and PMMoV) were designed manually for each virus, and the best primers standardized for multiplex PCR conditions are shown in Table 1. All the primer sets were synthesized at a commercial facility (GCC Biotech India Pvt. Ltd., West Bengal, India).

### 2.4. Singleplex PCR and Its Optimization

Initially, the virus-specific primers were used to detect the respective virus in singleplex PCR using 50 ng of cDNA from CaCV, ChiVMV, LCCV, CMV, and PMMoV and DNA from ChiLCV positive samples, employing the procedure mentioned in Section 2.2. A total of 25 μL PCR reaction volume included 0.13 μL (5 u/µL) of GoTaq^®^, taq polymerase; 5 μL of 5× GoTaq^®^ Green Buffer (Promega, Madison, WI, USA); 1 μL dNTP (10 mM); and 1 μL of forward and reverse primer mixtures (10 µM), and the final volume was adjusted to 25 μL with nuclease-free water. The PCR procedure consisted of an initial denaturation step at 95 °C for 5 min, followed by 35 cycles of denaturation at 94 °C for 1 min, annealing at 50 °C for 50 s, extension at 72 °C for 1.20 min, and final extension at 72 °C for 10 min. The amplicons were separated on 2% agarose gels (ethidium bromide dyed) in a 1× TAE buffer and visualized using the Gel Doc XR system (Bio-Rad, Hercules, CA, United States) and sequenced bi-directionally at a commercial facility (Eurofins Genomics India Pvt. Ltd., Bengaluru, India). In all the singleplex experiments, plasmid DNA containing a viral gene insert of respective viruses and the cDNA/DNA of an infected sample were used as a positive control, and the cDNA/DNA of a healthy sample was used as a negative control. Water was used as a non-template control.

### 2.5. Development of Multiplex PCR and Its Optimization

Multiplex PCR was optimized to enable the synchronous amplification of six target regions in one reaction. To ensure optimal amplification conditions for the six target sequences in a multiplex PCR assay, a range of PCR-related parameters were evaluated, such as the annealing temperatures of the primers, extension time, and the concentrations of primers. An equal concentration (50 ng) of each of the plasmid DNA containing the gene inserts of CaCV, ChiVMV, ChiLCV, LCCV, CMV, and PMMoV were mixed as templates for multiplex PCR. The optimized factors of the multiplex PCR were the ratio of the 6 pairs of primers with equal and unequal concentrations, annealing temperature (46–56 °C, in 2 °C increments), and extension time (1–2.5 min). Multiplex PCR reactions were amplified using 0.25 μL (5 u/µL) of GoTaq^®^, taq polymerase 5 μL of 5× GoTaq^®^ Green Buffer (Promega, Madison, WI, USA); 2.5 μL MgCl_2_ (25 mM); 2 μL dNTP (10 mM); 6.7 μL of total forward and total reverse primer mixtures (0.675 μL, CaCVmCF/R; 0.475 μL, ChiVMVmF/R; 0.575 μL, Chirke-mF/R; 0.675 μL, LCVmF/R; 0.475 μL, CMVmF/R and 0.475 μL, PMMoVmCF/R; 10 µM); and 50 ng of mixed plasmid DNA containing respective viral gene inserts as template, and the final volume was adjusted to 25 μL with nuclease-free water. In multiplex PCR experiments, a mixture of plasmid DNA-containing viral gene insert of respective viruses (50 ng) was used as a positive control and the cDNA/DNA of a healthy sample was used as negative control. Water was used as a non-template control.

### 2.6. Sensitivity Determination of Multiplex PCR

Purified plasmid DNA containing the gene inserts of CaCV, ChiVMV, ChiLCV, LCCV, CMV, and PMMoV were serially diluted using 10-fold serial dilution (10^−1^–10^−9^) from 50 ng/μL to 0.50 fg/µL, respectively. Then, 1 μL of each dilution was used as a template in a sensitivity evaluation of the multiplex PCR assay. This assay was performed to determine the limit of detection (i.e., the minimum concentration of plasmid DNA containing the respective viral genes that can be detected by the developed multiplex PCR method) [58].

### 2.7. Evaluation of the Stability and Validation of the Multiplex PCR Using Field Samples

To rule out the possibility of cross-reactions, the multiplex PCR was evaluated by removing a primer pair at a time from the total multiplex reaction mixture of the multiplex PCR. Respective primer pairs of CaCV, ChiVMV, ChiLCV, LCCV, CMV, and PMMoV were removed individually from the total master mix solution while the other five primer pairs were added in the mixtures to assess the multiplex PCR system for stability, and the results were analyzed.

Symptomatic and asymptomatic chilli samples (*C. annum*, *C. chinense*, and *C. frutescens*) were collected from different fields of Northeast India. Both the DNA and RNA were extracted from all the samples to capture both the virus groups. The total RNAs were extracted with a RNeasy^®^ Plant mini kit (Qiagen, Hilden, Germany), and 50 ng of the total RNA was used for synthesizing the first-strand cDNA using an M-MLV Reverse Transcriptase kit (Promega, Madison, WI, USA). The total DNA was extracted using a DNeasy^®^ Plant Mini Kit (Qiagen, Hilden, Germany) following manufacturers protocol. An equal concentration of 50 ng each of the cDNA and DNA were taken and mixed properly. Then, 6 μL (50 ng/μL) of cDNA and DNA mixture was used as a template and processed for multiplex PCR protocols. In all the validation experiments, a mixture of plasmid containing viral gene inserts of all six viruses was used as a positive control, the cDNA and DNA mixture of a healthy plant as a negative control, and water as a non-template control.

## 3. Results

### 3.1. Singleplex PCR and Its Optimization

In this study, the virus-specific primers individually designed from the conserved region of the viral genome (best working primer pairs listed in Table 1) were initially tested for the specificity of employing cDNA, DNA, and plasmid DNA containing viral gene inserts (50 ng each) as a template using temperature-gradient PCR at six different annealing temperatures (46 °C, 48 °C, 50 °C, 52 °C, 54 °C, and 56 °C), which yielded specific amplicons of 150 bp, 250 bp, 350 bp, 450 bp, 572 bp, and 634 bp size corresponding to the expected target sequences of CaCV, ChiVMV, ChiLCV, LCCV, CMV, and PMMoV, respectively, at all annealing temperatures between 46–56 °C (Figure 1A–F). The specific amplicons obtained in a singleplex PCR assay were cloned and sequenced. The sequence comparison with viral sequences in the NCBI database showed high homology (>99.6%) with the respective virus genome reference, which confirmed their viral origin.

### 3.2. Development of Multiplex PCR and Its Optimization

The primers that gave specific amplification in singleplex PCR were further subjected to the optimization of multiplex conditions using plasmid DNA containing viral gene inserts (50 ng each) in multiplex PCR for the simultaneous detection of all six target viruses in a single assay. The specific amplification of respective targets was obtained at annealing temperatures between 48 °C and 56 °C (Figure 1G). Based on the brightness of the target amplification band as observed on agarose gel electrophoresis, the annealing temperature of 50 °C was considered optimal and used in the subsequent multiplex PCR experiments.

The extension time for the singleplex and multiplex PCR was also optimized. The optimum extension time was based on the size of the largest amplicon in the mPCR reaction (i.e., PMMoV). Extension periods of less than 1 min did not give the amplification of all the six targets, but an increase in the duration to 1.20 min gave the amplification of all six targets, thus it was considered the most suitable extension time for the mPCR set-up.

To enable the selection of primer concentrations that gave similar amplification of each target, multiplex PCR was carried out using primers with the same (Figure 2A–F, Figure 2G lane 1–6) as well as different (Figure 2G, lane M1, M2, and M3) concentrations using the plasmid DNA containing gene inserts of each virus (50 ng each). ChiVMV, CMV, and PMMoV were preferentially amplified with any of the primer concentrations used, both in singleplex and multiplex conditions (Figure 2G).

Therefore, PCR conditions were optimized using the primer concentrations of 450 nM for CaCVmCF/R, 150 nM for ChiVMVmF/R, 350 nM for LCVmF/R, 150 nM for Chirke-mF/R, 150 nM for CMVmF/R, and 150 nM for PMMoVmCF/R (Figure 2G, lane M2). This indicated that reducing the primer concentrations that amplify preferentially over others (i.e., the primer pairs of ChiVMV, LCCV, CMV, and PMMoV) and increasing the primer concentrations that amplify least (i.e., the primer pairs of CaCV and ChiLCV) gave a more suitable result than using the same concentration for all the primer pairs.

The results also showed the amplification of CaCV, ChiLCV and LCCV in multiplex PCR was more efficient than in singleplex PCR while ChiVMV, CMV, and PMMoV were preferentially amplified (Figure 1G, Figure 2G and Figure 3G).

### 3.3. Sensitivity Determination of the Multiplex PCR

To evaluate the sensitivity of the developed multiplex PCR method, 50 ng of plasmid DNA (containing viral gene inserts) of CaCV, ChiVMV, ChiLCV, LCCV, CMV, and PMMoV were serially diluted and used as templates. The targeted fragments could be amplified at the lowest concentration of 500 fg/µL for CaCV and ChiVMV, 5.00 fg/µL for ChiLCV and LCCV, and 50.00 fg/µL for CMV and PMMoV (Figure 3A–F). Additionally, the minimum amount of the virus nucleic acid that could be detected using the multiplex PCR method was also determined likewise, suggesting 5.00 pg/µL of plasmid DNA containing viral inserts was needed to detect all of the viruses simultaneously in multiplex PCR (Figure 3G). These results showed that the developed multiplex PCR assay had sufficient sensitivity for use in the routine diagnosis purposes.

### 3.4. Determination of Stability of Multiplex PCR

Despite having the mixture of six plasmid DNA (containing gene inserts of six viruses) used as template, only the target specific amplification to the added primer set was observed. When a specific primer pair was removed purposely, there was no visible band of that specific reaction (Figure 4). Primer optimization using temperature gradient PCR showed that the Tm of the primers ranged from 49 °C to 54 °C (Figure 1), suggesting that a common annealing temperature was possible. The PCR result also showed that the amplified product ranged from 150 to 634 bp correspondingly in size to the expected target sequence from CaCV, ChiVMV, ChiLCV, LCCV, CMV, and PMMoV, respectively. These all demonstrated the stability and specificity of the designed primers and suggested they could be successfully incorporated in a multiplex PCR and could be easily distinguished using standard agarose gel electrophoresis.

### 3.5. Detection of Multiple Viruses in Chilli Collected from the Fields Using Multiplex PCR

Sixty-one chilli leaf samples exhibiting the symptoms of mottling, yellow-mosaic, leaf curling, chlorotic, and necrotic spots were collected from different locations of Northeast India, covering the states of Arunachal Pradesh, Manipur, Nagaland, Meghalaya, and Sikkim, and used for validation of developed multiplex PCR assay (Table 2). The samples were assayed by the developed multiplex PCR, and the results showed that samples often had mixed virus infections (the mPCR assay for the representative samples is shown in Figure 5). Out of the total samples, 55 were positive for either of the virus; 37 were tested positive for CaCV, 31 for ChiVMV, 17 for ChiLCV, 22 for LCCV, 35 for CMV, 18 for PMMoV, and 7 had a mixed infection of all the viruses (PMMoV, CaCV, ChiVMV, LCCV, and CMV). CaCV, ChiVMV, LCCV, CMV, and PMMoV were found to be more prevalent (Table 2). Interestingly, only 22 samples (36%) were positive for a single virus infection; however, 33 samples (54%) were positive for three or more viruses tested in mPCR, showing a widespread occurrence of mixed infection under field conditions. The virus-specific amplicons of 26 mPCR positive samples for either of viruses (out of total 55 positive samples) were subjected to sequencing and sequencing results revealed their exact viral identity and robustness of developed assay.

## 4. Discussion

The general approaches for plant virus detection are serological and molecular methods. Serological methods, such as ELISA, can handle a large number of samples over a short period of time despite limitations like low sensitivity, numerous steps involved, and ability to detect individual virus at a time. In addition, the benchmark PCR and RT-PCR assays are time consuming and cost expensive. The crop plants are often infected with multiple viruses belonging to different taxonomic groups. The crops like chilli are susceptible to multiple viruses and therefore, mixed infections are a common occurrence. Multiple viruses are responsible for the rapid destruction of chilli crops in India; however, its impact on chilli growth and yield has not been established properly. The diversity in symptoms and overlapping symptomatic expressions caused by different viruses make the symptom-based detection unreliable and tedious. Moreover, singleplex PCR assays require separate amplification of each virus of interest and are considerably resource intensive and expensive; therefore, a mPCR to detect multiple viruses—CaCV, ChiLCV, ChiVMV, CMV, LCCV, and PMMoV—in a single reaction is desirable. Further, it is of paramount importance to have the exact identity of virus infection identified and detected with the highest possible level of sensitivity. The development and optimization of multiplex PCR assays are of utmost importance in crops plants as infection of multiple viruses occurs more often.

The development of a multiplex PCR assay is often complex since primers must comply with several similar conditions that all primers can function under the same PCR conditions like compatibility, avoiding cross-binding, competition, and sufficient sensitivity [56]. Multiplex PCR has been successfully employed earlier to identify multiple viruses in different crops plants such as tomato [59], chilli/pepper [60], and tobacco [61]. Chilli is an important spice crop of India that is grown in almost all geographical areas. The infection of multiple RNA and DNA viruses has become an issue of the utmost importance to be addressed to achieve the successful production of chilli crop. The infection of ChiLCV, ChiVMV, CMV, PMMoV, and CaCV are the most devastating and of common occurrence under Indian conditions. Recently, the infection of LCCV in chillies grown in Northeast India was reported [21]. In this study, we report on the development, optimization, and validation of a multiplex PCR assay for the simultaneous detection of five RNA viruses (ChiVMV, CMV, PMMoV, CaCV, and LCCV) and a DNA virus (ChiLCV) which are of the most predominant occurrence and a major threat to chilli cultivation in India. Earlier, a multiplex RT-PCR assay for the simultaneous detection of CMV, tobacco mosaic virus, PMMoV, PVY, and TSWV in chilli was reported [60]; however, to the best of our knowledge, there is no report on multiplex PCR assay for the detection of six RNA and DNA viruses, which was attempted in the present study. We designed the primers to target the conserved regions of the viral genome in order to achieve specific amplification and avoid any chances of cross-reactivity. The specific primers were initially tested and validated in the singleplex followed by multiplex PCR assays. The standardization of different parameters of multiplex PCR assay was done in order to develop a sensitive and robust assay for the simultaneous detection of six target viruses in a single reaction tube. The annealing temperature is generally considered the critical factor for the specificity and amplification efficiency of PCR [62,63]. The analysis of the virus-specific primers in this study showed that the melting temperature (Tm) of all the primers were sufficiently close (49–54 °C) to allow a common annealing temperature. This was confirmed experimentally, and the annealing at 50 °C allowed for an optimum amplification of all the viral targets efficiently. The primer concentration is one of the most important factors affecting amplification efficiency in multiplex PCR, where numerous primers and templates are present [64,65,66]. Therefore, the optimization of primer concentrations is important to achieve the best amplification for all targets efficiently. The use of an equal primer concentration was reported to amplify some targets preferentially over others [66]. Therefore, reducing the primer concentration of those targets that were amplified more preferentially (i.e., ChiVMV, LCCV, CMV and PMMoV primers) compared to those that were repressed (i.e., CaCV and ChiLCV primers) in the multiplex PCR with differential concentration of each primer set (450, 150, 350, 150, 150, 150 nM for CaCV, ChiVMV, ChiLCV, LCCV, CMV and PMMoV respectively) enabled an optimal amplification of all six target viruses in the mPCR assay. The most commonly used extension temperature in PCR is 72 °C because it is considered the optimum temperature for Taq DNA polymerase, and the optimization of extension temperatures has rarely been reported. Some reports have shown the improved amplification efficiency in the case of multiplex PCR at an extension temperature lower than 72 °C [67,68,69]. A lower extension temperature reduces primer interaction and, therefore, increases the amplification efficiency in multiplex PCR [66]. The standardization of extension temperatures of 60, 63, 66, 69, 72, and 75 °C in the developed multiplex PCR in the present study showed that all the tested temperatures showed similar efficiency for all the targets, both for singleplex and multiplex PCR, which is contradictory to previous reports. The newly-developed multiplex PCR for chilli viruses was found to be a reliable and sensitive method for the simultaneous detection of six chilli viruses irrespective of their genetic composition in the amplification of single and multiple targets. The developed mPCR assay was field validated using a set of 61 symptomatic and asymptomatic chilli samples. In them, 90% of the tested samples had an infection of either type of virus, and 54% were infected with three or more number of viruses. This clearly demonstrated the widespread prevalence of mixed virus infection in chilli and the practical utility of developed mPCR assay in the routine detection of chilli viruses. Six samples were negative for all viruses, out of which two were asymptomatic. Of the remaining four, some showed distortion and curling symptoms but tested negative for all the viruses. This could be attributed to the infection of non-targeted viruses or other biotic agents. To the best of our knowledge, it is the first report of a multiplex PCR system that can simultaneously detect both RNA and DNA viruses infecting chilli. Considering the convenience and high efficiency of this method, it will be suitable for the detection, epidemiological investigation, and providing of the clues on key traits that could eventually provide a wide range of tolerances to multiple viral stresses or for the development of other multiplex PCR assays for the detection of pathogens in plants.

## Figures and Tables

**Figure 1 pathogens-12-00006-f001:**
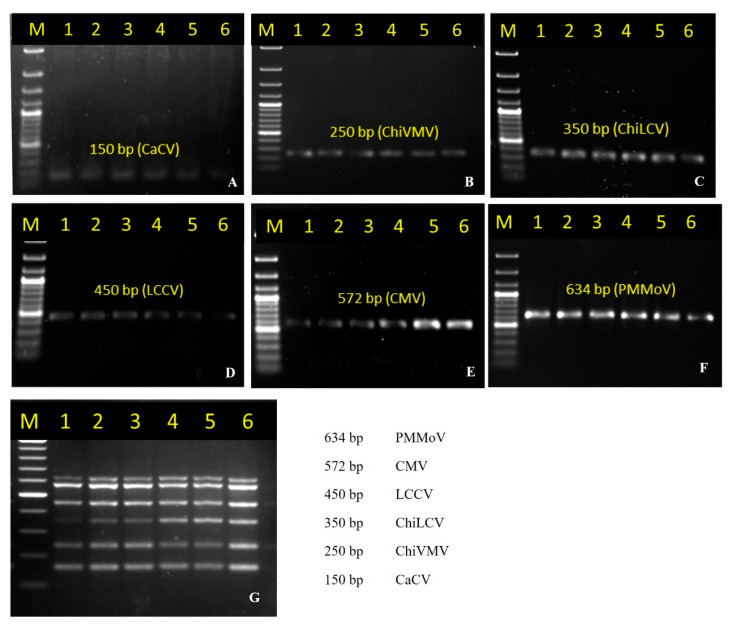
Optimization of annealing temperatures. The multiplex PCR was optimized by performing primer annealing-temperature gradient-PCR (lane 1–6) at 46 °C, 48 °C, 50 °C, 52 °C, 54 °C and 56 °C respectively, using plasmid DNA containing the gene inserts of CaCV, ChiVMV, ChiLCV, LCCV, CMV and PMMoV respectively. Lane M; 100 bp DNA ladder (GCC Biotech India, India). (**A**) PCR using primers CaCVmCF/R amplifying 150 bp; (**B**) PCR using primers ChiVMVmF/R amplifying 250 bp; (**C**) PCR using primers LCVmF/R amplifying 350 bp (**D**) PCR using primers Chirke-mF/R amplifying 450 bp; (**E**) PCR using primers CMVmF/R amplifying 572 bp (**F**) PCR using primers PMMoVmCF/R amplifying 634 bp (**G**) Multiplex PCR using 6 sets of primers at 6 different annealing temperatures.

**Figure 2 pathogens-12-00006-f002:**
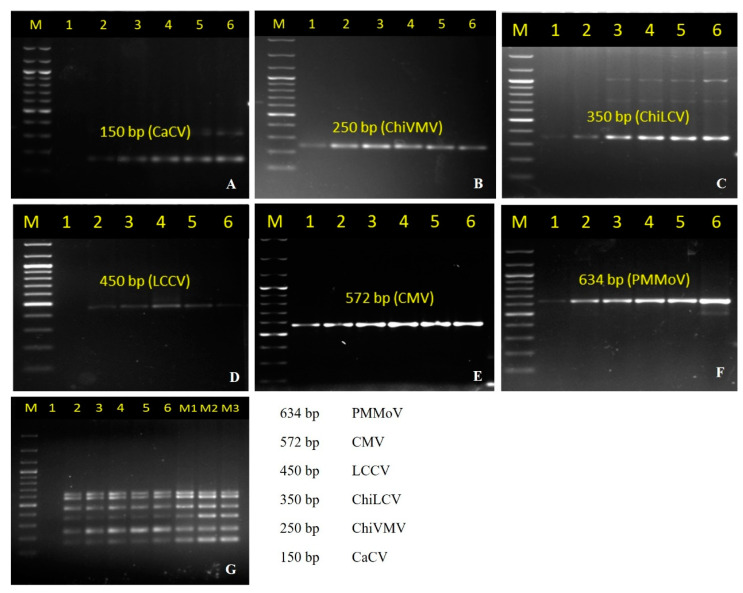
Optimization of primer concentrations (50–550 nM) for singleplex and multiplex PCR. The mPCR primer concentration were optimized using different as well as same concentrations by using the plasmid DNA containing the respective viral gene inserts. Lane 1–6: (50, 150, 250, 350, 450 and 550 nM respectively) with equivalent primer concentration for all the six primer pairs used. (**A**) Different concentration of primer pair CaCVmCF/R (CaCV); (**B**) Different concentration of primer pair ChiVMVmF/R (ChiVMV); (**C**) Different concentration of primer pair LCVmF/R (ChiLCV); (**D**) Different concentration of primer pair Chirke-mF/R (LCCV); (**E**) Different concentration of primer pair CMVmF/R (CMV); (**F**). Different concentration of primer pair PMMoVmCF/R (PMMoV) (**G**) Multiplex PCR using equal primer concentration (lanes 1–6), and unequal primer pair concentrations, Lane M1: CaCVmCF/R = 350 nM, ChiVMVmF/R = 150 nM, LCVmF/R = 150 nM, Chirke-mF/R = 150 nM, CMVmF/R = 150 nM and PMMoVmCF/R = 150 nM; M2: CaCVmCF/R = 450 nM, ChiVMVmF/R = 150 nM, LCVmF/R = 350 nM, Chirke-mF/R = 150 nM, CMVmF/R = 150 nM and PMMoVmCF/R = 150 nM; M3: CaCVmCF/R = 550 nM, ChiVMVmF/R = 150 nM, LCVmF/R = 550 nM, Chirke-mF/R = 150 nM, CMVmF/R = 150 nM and PMMoVmCF/R = 150 nM. Lane M: 100-bp DNA ladder (GCC Biotech India, India).

**Figure 3 pathogens-12-00006-f003:**
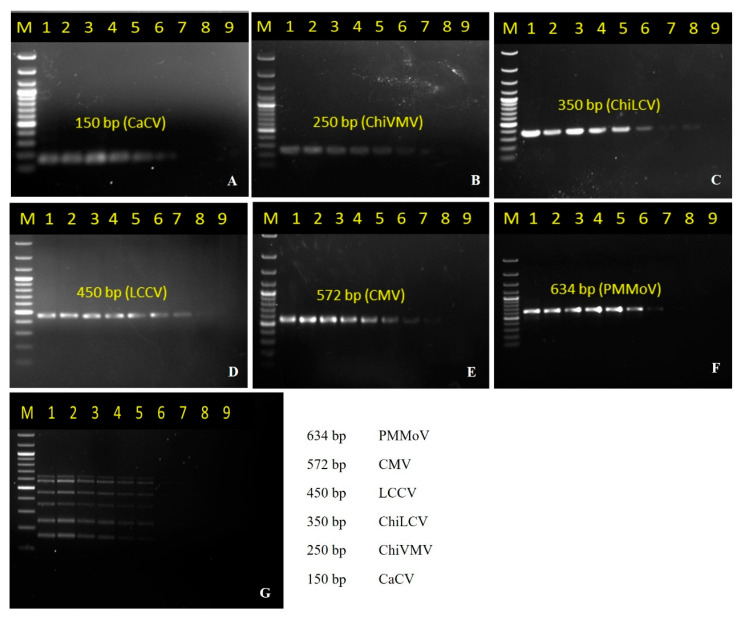
Evaluation of the detection limit of singleplex and multiplex PCR using serially diluted individual and mixed plasmid DNA containing viral inserts. The detection limit of CaCV, ChiVMV, ChiLCV, LCCV, CMV and PMMoV, were performed individually as well as in multiplex PCR, as represented by the lanes 1–9. The concentrations used in lane 1–9 respectively are 50 ng/µL, 5 ng/µL, 500 pg/µL, 50 pg/µL, 5 pg/µL, 500 fg/µL, 50 fg/µL, 5 fg/µL and 0.5 fg/µL. Lane M: 100 bp DNA ladder (GCC Biotech India, India). (**A**) PCR using primer CaCVmCF/R amplifying 150 bp; (**B**) PCR using primer ChiVMVmF/R amplifying 250 bp; (**C**) PCR using Primer LCVmF/R amplifying 350 bp (**D**) PCR using primer Chirke-mF/R amplifying 450 bp; (**E**) PCR using primer CMVmF/R amplifying 572 bp (**F**) PCR using primer PMMoVmCF/R amplifying 634 bp (**G**) Multiplex PCR using 6 sets of primers and serially diluted individual and mixed plasmid DNA for evaluation of the detection limit.

**Figure 4 pathogens-12-00006-f004:**
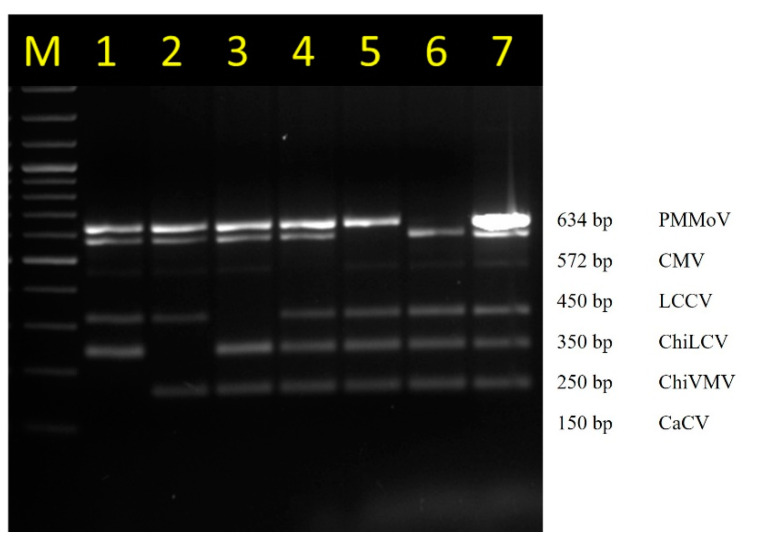
Specificity of the multiplex PCR. The specificity was confirmed by the absence of amplification of CaCV, ChiVMV, ChiLCV, LCCV, CMV and PMMoV amplicons in the lane 1–6 respectively when their respective primer pairs were removed from the total mixtures of all the plasmid DNA containing viral gene inserts as template. Lane 7: Multiplex PCR with all the six primer pairs. Lane M: 100 bp DNA ladder (GCC Biotech India, India).

**Figure 5 pathogens-12-00006-f005:**
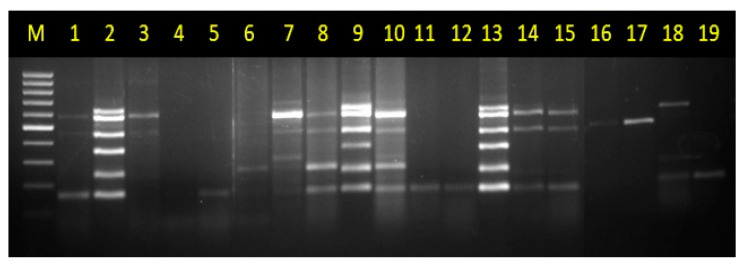
Validation of the multiplex PCR. The developed mPCR assay was validated using chilli samples collected from field, the DNA and RNA of representative 19 chilli samples (sample loaded in lane 1 to 19 are sample number 18; 12; 3; 4; 5; 7; 1; 10; 16; 13; 48; 58; 53; 49; 6; 19; 20; 61; 27 respectively as listed in Table 2) collected from different locations of North East India were tested using the developed multiplex PCR assay. Lane M: 100 bp DNA ladder (GCC Biotech India, India).

**Table 1 pathogens-12-00006-t001:** The virus-specific primers used in the multiplex PCR assay.

Primer	Sequence (5′–3′)	Target Virus	Length (bp)	Tm (°C)	Amplicon Size (bp)	Primer Binding Site
CaCVmCF	GGGAAAATAATTGGTGCAAGG	CaCV	21	57.5	150	317–337
CaCVmCR	AGATGTATGCAAAGGTAATGGAATT		25	59.2		473–449
ChiVMVmF	AGCGCAACTCTGAGAAGCC	ChiVMV	19	59.5	250	9141–9159
ChiVMVmR	TTCTATTCACATCCTCTGCTG		21	57.5		9392–9372
LCVmF	CCCATAGAGTAGGTAAGCG	ChiLCV	19	57.5	350	610–628
LCVmR	CATATTTACCAGCTTCCTGC		20	56.4		964–945
Chirke-mF	GTGGAGAATACCTCAAATACC	LCCV	21	57.5	450	117–137
Chirke-mR	CTGAGTATACCCTCTTTTTGTG		22	58.4		573–552
CMVmF	ATGGACAAATCTGAATCAACC	CMV	21	55.4	572	1–21
CMVmR	GTCTTTTGAATACACGAGTAC		21	55.4		573–553
PMMoVmCF	AAAGGAAGTAATAAGTATGTAGGTAAGAG	PMMoV	29	63.3	634	5551–5579
PMMoVmCR	GTTCGTCCAACTTATTTATGCC		23	58.4		6184–6163

Primer binding site with respect to the GenBank sequences MN707994, KR296797, MN417111, MH899147, MK652151, MN267900 for CaCV, ChiVMV, ChiLCV, LCCV, CMV and PMMoV respectively.

**Table 2 pathogens-12-00006-t002:** Validation of multiplex PCR assay using the field samples of chilli collected from different regions of North East India.

Sample No.	State	Location	Species of Chilli	Symptoms	Virus Detection					
					CaCV	ChiVMV	ChiLCV	LCCV	CMV	PMMoV
1	Manipur	Chandrakhong	King chilli (*C. chinense)*	Mos, Nec, Wst	+	−	−	+	+	−
2	Manipur	Khoijuman	King chilli (*C. chinense)*	Chl, Nec, Mos, Stn	+	+	+	+	+	+
3	Tripura	Khwai	Chilli (*C. annuum*)	Mos, Ss	−	−	−	−	+	−
4	Mizoram	Kolasib	King chilli (*C. chinense*)	Dis	−	−	−	−	−	−
5	Manipur	Mongsang, Tamenglong	Bird’s eye chilli (C. *frutescens*)	Mos, Nec, Chl	+	−	−	−	−	−
6	Manipur	Tamenglong	Chilli (*C. annuum*)	Mos, Nec, Chl	+	−	−	+	+	−
7	Manipur	Gelnal, Kangpokpi	King chilli (*C. chinense)*	Ss, Mot, Dis	−	+	−	−	−	−
8	Manipur	Gelnal, Kangpokpi	Chilli (*C. annuum*)	Ss, Mot, Dis	−	+	−	−	−	−
9	Manipur	Maklode, Senapati	Chilli (*C. annuum*)	Dis, Mot, Mos, Chl, Nec, Stn	+	−	+	+	+	+
10	Manipur	Tupul, Tamenglong	Bird’s eye chilli (C. *frutescens*)	Mos, Ss, Cur	+	+	−	+	+	−
11	Manipur	Mongsang, Tamenglong	Chilli (*C. annuum*)	Mos, Ss, Cur	+	+	−	+	+	−
12	Manipur	Sirarakhong, Ukhrul	King chilli (*C. chinense)*	Mos, Stn, Wst	+	+	+	+	+	+
13	Manipur	Purul, Senapati	King chilli (*C. chinense)*	Dis, Ss, Chl, Nec, Stn	+	+	−	+	+	−
14	Manipur	Sawombung	King chilli (*C. chinense)*	Dis, Mos, Nec	+	−	−	−	−	−
15	Manipur	Kachai, Ukhrul	King chilli (*C. chinense)*	Mos, Chl, Nec, Stn	+	−	-	−	−	−
16	Manipur	Hiyangthang	King chilli (*C. chinense)*	Chl, Mos, Stn, Ss	+	+	+	+	+	+
17	Sikkim	Lingi Payong	Dale chilli (*C. annuum)*	Mot, Nec, Wst	+	+	−	+	+	−
18	Sikkim	Rishi	King chilli, *C. chinense*	Dis, Stn, Wst	+	-	-	+	+	-
19	Arunachal Pradesh	Pasighat	Chilli (*C. annuum*)	Chl, Stn	−	−	−	+	−	−
20	Manipur	Tera, Imphal	Black chilli (*C. annuum*)	Wst	−	−	−	+	−	−
21	Nagaland	Wokha	King chilli (*C. chinense*)	Dis, Mos, Cur	−	+	+	−	−	+
22	Manipur	Nambol kakyai	King chilli (*C. chinense*)	Dis, Mot, Mos, Ss	−	+	−	−	−	−
23	Manipur	Tuisen, Ukhrul	King chilli *(C. chinense)*	Mos, Chl, Nec, Stn	+	+	−	-	+	+
24	Mizoram	Thenzawl	King chilli (*C. chinense*)	Mot, Mos, Ss, Cur, Chl, Nec, Stn,	+	+	+	−	+	+
25	Meghalaya	Barapani	King chilli (*C. chinense*)	Dis, Mot, Mos, Stn,	−	−	−	−	+	−
26	Manipur	Sekta	King chilli (*C. chinense*)	Dis	−	−	−	−	−	−
27	Manipur	Pungdongbam	King chilli (*C. chinense*)	Chl, Nec	+	−	−	−	−	−
28	Manipur	Sirarakhong	King chilli (*C. chinense*)	Mot, Ss, Stn	−	+	−	−	−	−
29	Manipur	Muolvaiphei	King chilli (*C. chinense*)	Dis, Mot, Mos, Ss, Chl, Stn,	+	−	+	−	+	+
30	Manipur	Wangoi	King chilli (*C. chinense*)	Mos, Cur, Chl, Stn	+	+	−	−	+	−
31	Manipur	Nambol	King chilli (*C. chinense*)	Mos, Ss, Cur, Chl, Nec, Stn	+	+	+	−	+	+
32	Manipur	Sinam Village	King chilli (*C. chinense*)	Dis, Mot, Mos, Ss, Cur, Chl, Nec	+	+	−	−	+	−
33	Manipur	Wari	King chilli (*C. chinense*)	Mot, Stn, Wst	+	+	+	−	+	+
34	Manipur	Paorabi	King chilli (*C. chinense*)	Mot, Mos	−	−	−	−	+	−
35	Manipur	Sagolmang	King chilli *(C. chinense)*	As	−	−	−	−	−	−
36	Manipur	Sagolmang	Bird’s eye chilli (*C. fructescens*)	As	−	−	−	−	−	−
37	Manipur	Buhsau	King chilli (*C. chinense*)	Chl, Nec	+	−	−	−	−	−
38	Manipur	Zeezaw	King chilli (*C. chinense*)	Mot, Ss	−	+	−	−	−	−
39	Manipur	P. Sejol	King chilli (*C. chinensis*)	Dis, Mos, Cur, Chl	+	−	+	+	+	+
40	Manipur	P. Sejol	Bird’s eye chilli (*C. fructescens*)	Dis, Mos, Cur, Chl	+	−	+	+	+	+
41	Manipur	Saiton Khunou	King chilli (*C*. *chinense*)	Mot, Mos, Cur, Chl, Ne	+	+	−	+	+	−
42	Manipur	Khurai Angom Leikai	King chilli (*C. chinense*)	Mot, Nec, Stn, Wst	+	+	+	+	+	+
43	Manipur	Uchiwa	King chilli (*C. chinense*)	Dis, Mot, Mos	+	+	−	+	+	−
44	Manipur	Mayang Imphal	Chilli (*C. annuum*)	Chl, Stn, Wst	+	+	+	+	+	+
45	Manipur	Heibongpokpi	King chilli (*C. chinense*)	Dis, Mot, Mos	−	−	−	−	+	−
46	Manipur	Heibongpokpi	Chilli (*C. annuum*)	Dis, Mot, Mos	−	−	−	−	+	−
47	Manipur	Haorangsabal	King chilli *(C. chinense)*	Cur	−	−	−	−	−	−
48	Manipur	Lamsang	King chilli *(C. chinense)*	Chl, Stn	+	−	−	−	−	−
49	Manipur	Lambal	King chilli *(C. chinense)*	Dis, Mot, Ss	+	−	−	+	+	−
50	Manipur	Lambal	Bird’s eye chilli (*C. fructescens*)	Dis, Mot, Ss	−	+	−	−	−	−
51	Manipur	Songpijang	King chilli (*C. chinense*)	Dis, Mot, Mos, Chl	+	−	+	+	+	+
52	Manipur	Gelnel	King chilli (*C. chinense*)	Stn, Wst, Mot	+	+	−	+	+	−
53	Manipur	Chlava	King chilli (*C. chinense*)	Mot, Mos, Ss, Stn	+	+	+	+	+	+
54	Manipur	Rengpang	King chilli (*C. chinense*)	Dis, Mot, Mos, Ss, Chl	+	+	−	+	+	−
55	Manipur	Gopibung	King chilli (*C. chinense*)	Mot, Mos, Chl, Stn	+	+	+	+	+	+
56	Manipur	Khongshang	King chilli (*C. chinense*)	Dis, Mos, Stn	−	−	-	−	+	−
57	Nagaland	New chumoukedima	King chilli (*C. chinense*)	Dis	−	−	−	−	−	−
58	Nagaland	Medziphema	King chilli (*C. chinense*)	Dis, Chl, Stn	+	−	−	−	−	−
59	Nagaland	Old Tesen	King chilli (*C. chinense*)	Mot, Ss, Stn,	−	+	−	−	−	−
60	Nagaland	Duekwaram	King chilli (*C. chinense*)	Mos, Cur	+	−	+	−	+	+
61	Nagaland	Jharnapani	King chilli (*C. chinense*)	Dis, Mos, Chl, Nec	+	+	−	−	+	−
**Total positive sample**					**37**	**31**	**17**	**22**	**35**	**18**

Symptoms: As—Asymptomatic, Dis—leaf distortion, Mot—mottling, Mos—yellow mosaic, Ss—shoe-string, Cur—leaf curling, Chl—Chlorotic spot, Nec—Necrotic spot, Stn—Stunted, Wst—White streak. The results of Multiplex PCR assay were in-agreement with the results of singleplex PCR assay of each sample.

## Data Availability

The datasets generated during and/or analysed during the current study are included in the manuscript and are available from the corresponding author on request.

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
