# Peer review of "A Simplified Multiplex PCR Assay for Simultaneous Detection of Six Viruses Infecting Diverse Chilli Species in India and Its Application in Field Diagnosis"

_pathogens, 2022, doi:10.3390/pathogens12010006_

Round 1

Reviewer 1 Report

The manuscript pathogens-2082409 (A Simplified Multiplex PCR Assay for Simultaneous Detection of Six Viruses Infecting Diverse Chilli Species in India and its Application in Field Diagnosis) provides information on the adjustment and implementation of the multiplex PCR technique for the diagnosis of six virus of chilis. The technique is important for those who work on this problem and in the area mentioned in the work. It is clearly and accurately written and, in my opinion, it is a work that could be published after the incorporation of minor revisions.

Author Response

The manuscript pathogens-2082409 (A Simplified Multiplex PCR Assay for Simultaneous Detection of Six Viruses Infecting Diverse Chilli Species in India and its Application in Field Diagnosis) provides information on the adjustment and implementation of the multiplex PCR technique for the diagnosis of six virus of chilis. The technique is important for those who work on this problem and in the area mentioned in the work. It is clearly and accurately written and, in my opinion, it is a work that could be published after the incorporation of minor revisions.

Response: We thank the Expert reviewer for highlighting the importance of work and appreciation. We also thank the reviewer for critical suggestions for its improvement. We have addressed all the comments in the revision and feel that manuscript have improved with respect to quality and presentation.

Comment 1: A diagnostic technique does not combat an infection, in any case it can help to establish management measures

Response: Revised in the view of Expert Reviewer's comment (please see abstract of revised manuscript)

Comment 2: Suggestion: green = fresh?

...globe [1,2], with a production of 4.25 million tonnes of both green and dry chilli peppers each year [3].

Response: Revised accordingly (please see Introduction section of revised manuscript)

Comment 3: each year?

Response: Changed to ‘annual’

Comment 4: The acronym of each virus was mentioned above. From here, always use the full name or acronym

Response: Changed accordingly (please see the revised manuscript)

Comment 5: Indicate how many primer pairs were tested for each virus

Response: As per suggestion, the information has been included in section 2.3 of revised manuscript.

Three sets of primers for ChiVMV, ChiLCV, CMV and four sets of primers for LCCV, CaCV, PMMoV were designed manually and best primer pair was taken forward for further studies.

Comment 6: Indicate the protocols for extracting nucleic acids, obtaining cDNAs, and the amount of cDNA and DNA used, etc.

Response: As advised by the Expert Reviewer, the details have been provided (please see section 2.7 of revised manuscript)

Both the DNA and RNA were extracted from all the samples to capture both the virus groups. Total RNAs were extracted with RNeasy® Plant mini kit (Qiagen, Hilden, Germany) and 50 ng of the total RNA was used for synthesizing the first-strand cDNA using M-MLV Reverse Transcriptase kit (Promega, Madison, WI, USA). Total DNA was also extracted using DNeasy® Plant Mini Kit (Qiagen, Hilden, Germany), following manufacturers protocol. Equal concentration of 50 ng each of the cDNA and DNA were taken and mixed properly. Then 50 ng of the mixture was used as the template to be assayed by the developed multiplex PCR. In all the validation experiments, mixture of plasmid containing viral gene inserts of all six viruses was used as positive control, cDNA and DNA mixture of healthy plant as negative control and water as non-template control.

Comment 7: Six samples were negative for all viruses. Were these samples symptomatic or asymptomatic? What are these results attributed to?

Response: The information on symptoms of all samples has been now included in the Table 2 now.

Six samples were negative for all viruses, out of which two were asymptomatic. Remaining four, some showed distortion and curling symptoms but tested negative for all the viruses. This could be attributed to the infection of non-targeted viruses or other biotic agents. (please see section 4 of revised manuscript)

Comment 8: Was DNA and RNA extracted from all the samples or depending on the virus in question?

Response: Both the DNA and RNA were extracted from all the samples since mixed infections of multiple viruses might occur naturally. The same has been included in the section 2.7 of revised manuscript

Comment 9: What primers were used for these PCR and RT-PCR?

Response: Primer pairs mentioned in Table were used for singleplex and multiplex PCR. The information is included in the revised manuscript.

Comment 10: Were the primers designed manually or using some software?

Response: The primers were designed manually (please see section 2.3 of revised manuscript)

Comment 11: Indicate which reactions were used as negative and positive controls in all assays performed. In results indicate as not shown.

Response:

In all the singleplex experiments, plasmid DNA containing viral gene insert of respective viruses and cDNA/DNA of infected sample were used as positive control, and cDNA/DNA of healthy sample was used as negative control. Water was used as non-template control.

In multiplex PCR experiments, mixture of plasmid DNA containing viral gene insert of respective viruses (50 ng) was used as positive control and cDNA/DNA of healthy sample was used as negative control. Water was used as non-template control.

(Please see sections 2.4 and 2.5 of revised manuscript)

Comment 12: Were the cDNA obtained in point 2.2 used here?

Reply: Yes, the same cDNAs obtained previously were used. It has been included in the revised manuscript (please see section 2.4).

Other changes as suggested in the annotated PDF file by Expert reviewer has been done (track changes of different colour fonts)

Reviewer 2 Report

Apart from grammatical issues, this manuscript is nicely laid out and the various parameters for experimental success were thoroughly tested. Very nice gel images. However, it would've been good to demonstrate that the assayed field samples harbored the correct viruses as I didn't notice anywhere in the manuscript that these samples were ever sequenced. Also, it would be nice to have an extra column in Table 2 stating whether the samples were from symptomatic vs. asymptomatic leaves. Multiple infections or different strains of a given virus(es) sometimes display reduced/no symptomatology.

Author Response

Response: We thank the Expert reviewer for highlighting the importance of work and appreciation. We have proof-read the manuscript and worked on grammatical issues.

The virus-specific amplicons of 26 mPCR positive samples for either of viruses (out of total 55 positive samples) were subjected to sequencing and sequencing results revealed their exact viral identity (please see section 3.5 of the revised manuscript).

The information on symptoms of all samples has been now included in the Table 2 now.

Reviewer 3 Report

In the article submitted a method for six pepper viruses detection based on multiplex PCR has been developed. Diagnostics of several plant viruses simultaneously, in one sample, are still relevant and necessary. The objectives of the work has been successfully solved. The PCR conditions (the primer concentration, the annealing temperature, the time and even the temperature of the extension step) were carefully selected. These experiments were performed using a model system based on recombinant plasmids containing inserted fragments of the virus genome. The results were validated in the field experiments. In the Introduction section all six viruses were characterized in detail, so the introduction look like a short review. This is very appropriate, since pepper viruses are not the most popular objects of research. I think the manuscript can be accepted for publication in the present form.

Author Response

We thank the Expert reviewer for highlighting the importance of work and appreciation.